# Effect of Storage on the Nutritional Quality of Queen Garnet Plum

**DOI:** 10.3390/foods10020352

**Published:** 2021-02-07

**Authors:** Gethmini Kodagoda, Hung T. Hong, Tim J. O’Hare, Yasmina Sultanbawa, Bruce Topp, Michael E. Netzel

**Affiliations:** 1Queensland Alliance for Agriculture and Food Innovation, The University of Queensland, Health and Food Sciences Precinct, Coopers Plains, QLD 4108, Australia; k.kodagoda@uq.edu.au (G.K.); h.trieu@uq.edu.au (H.T.H.); t.ohare@uq.edu.au (T.J.O.); y.sultanbawa@uq.edu.au (Y.S.); 2ARC Industrial Transformation Training Centre for Uniquely Australian Foods, Queensland Alliance for Agriculture and Food Innovation, The University of Queensland, Health and Food Sciences Precinct, Coopers Plains, QLD 4108, Australia; 3Queensland Alliance for Agriculture and Food Innovation, The University of Queensland, Maroochy Research Facility, Nambour, QLD 4560, Australia; b.topp@uq.edu.au

**Keywords:** Queen Garnet plum (QGP), domestic storage, plum tissues, bioactive compounds, phenolics, anthocyanins, carotenoids, sugars

## Abstract

Due to high perishability, plums are harvested at an early stage of maturity to extend postharvest storage life. Storage time and temperature can significantly affect the phytochemical and sugar composition of plums, altering their palatability and nutritional quality. In this study, variations in physiochemical properties (total soluble solids (TSS), titratable acidity (TA), color (chroma and hue angle)), phytochemical composition (total phenolic content (TPC), total anthocyanin content (TAC), and carotenoids), and sugars in three different tissues of the Queen Garnet plum (QGP) during storage at two common domestic storage temperatures (4 and 23 °C) were evaluated. There was an increase (*p* > 0.05) in TSS and a reduction (*p* < 0.05) in TA of the outer flesh at 23 °C. Chroma values of all the tissues reduced (*p* < 0.05) at 23 °C. At 4 °C, chroma values fluctuated between storage days. The TAC of the peel was the highest (*p* < 0.05) among the different tissues and continued to increase up to 10 days of storage at 23 °C (3-fold increase). At 4 °C, the highest (*p* < 0.05) TAC (peel) was observed after 14 days of storage (1.2-fold increase). TPC showed similar results. The highest (*p* < 0.05) TPC was recorded in the peel after 10 days of storage at 23 °C (2.3-fold increase) and after 14 days of storage at 4 °C (1.3-fold increase), respectively. Total carotenoids in the flesh samples at both storage temperatures were reduced (*p* < 0.05) after 14 days. Total sugars also decreased during storage. The results of the present study clearly showed that common domestic storage conditions can improve the nutritional quality of plums by increasing the content of bioactive anthocyanins and other phenolic compounds. However, the increase in phytochemicals needs to be counterbalanced with the decrease in total sugars and TA potentially affecting the sensory attributes of the plums.

## 1. Introduction

In recent years, the role of dietary phytochemicals in human health has become a major topic of discussion for nutritionists as well as the general public. The consumption of fruits and vegetables rich in phytochemicals, dietary fiber, and vitamins has been associated with a reduced risk in chronic diseases such as type 2 diabetes, hypertension, and cancers [1,2,3,4].

Plums are a popular stone fruit, belonging to the genus *Prunus*, and they are widely consumed around the world due to their attractive appearance, flavor, and aroma. Japanese blood plums (*P. salicina* Lindl.) are a good source of dietary phytochemicals such as phenolics (especially anthocyanins) [5,6], vitamins [7], and dietary fiber [8]. However, significant variations in physicochemical properties as well as phytochemical and nutritional composition are observed in plums depending on the cultivar, maturity stage, orchard practices, pre- and post-harvest treatments, and storage conditions [9,10,11,12].

Queen Garnet plum (QGP) is a Japanese blood plum cultivar which was developed as a high anthocyanin plum in a Queensland Government breeding program [13]. Studies have shown that consumption of QGP can reduce plasma and urine malondialdehyde (biomarker of oxidative stress) in healthy humans [14], reverse complications of metabolic syndrome in diet-induced obese and hypertensive rats [15], ameliorate symptoms of inflammatory bowel disease [16], reduce ambulatory blood pressure in younger and older adults [17], and inhibit platelet aggregation (anti-thrombotic activity) in healthy humans [18]. Plums are usually harvested at an early maturity stage and stored at low temperature (0-5 °C) for extending their postharvest shelf life, as they are highly perishable [19,20]. However, prolonged cold storage may also cause physiological changes that can reduce the shelf life and consumer acceptance of plums [12,20]. Therefore, careful selection of storage temperature and storage time is crucial in maintaining the nutritional quality of plums as long as possible. Due to the short harvest season, QGP are harvested and stored at 0–2 °C initially for several weeks depending on the initial fruit quality prior to retail sale. The aim of this study is to determine the impact of common domestic storage temperatures (4 °C and 23 °C) on (i) physicochemical properties, (ii) phytochemicals (total phenolics, anthocyanins and carotenoids), and (iii) sugar composition in different tissues of the QGP during a 14-day storage period.

## 2. Materials and Methods

### 2.1. Materials

Mature ripe QGP fruits were obtained from a commercial grower in Queensland, Australia in the 2018/2019 season. Mature fruits were harvested and preliminarily stored at 2 °C for 3 weeks (as a simulated pre-storage). After 3 weeks, approximately 200 plums were transported to the laboratory at Coopers Plains, Queensland, Australia and 90 homogeneous fruits (size, color and absence of any defect) were subsequently selected. The fruits were randomly divided into 9 groups of 10 fruits. Four groups were stored in a cold room at 4 °C, another 4 groups at room temperature (23 °C), and one group was used as reference for day 0. On 0, 4, 7, 10 and 14 days of storage, a group of fruit was taken randomly from each storage temperature. Each fruit was weighed and cut along a suture into two halves, and the peel was manually separated from the flesh. The flesh was further separated into outer flesh (OF; 7mm of flesh from the outer edge), and the inner flesh (IF) (Figure 1). The weight of each fruit tissue was recorded. The IF and OF were separately pureed (MM400 Retsch Mixer Mill, Haan, Germany), and a portion was used to measure chroma, hue angle, total soluble solids (TSS) and titratable acidity (TA). The separated tissues were freeze-dried and ground into powder and stored at −20 °C for further analysis.

Commercial standards of cyanidin-3-glucoside (C3G), sorbitol, glucose, fructose, sucrose, lutein, β-carotene, gallic acid, and all other chemicals and solvents (high performance liquid chromatography (HPLC) grade) were purchased from Sigma-Aldrich (Sydney, NSW, Australia) unless otherwise stated.

### 2.2. Determination of Physicochemical Properties

#### 2.2.1. Color Measurement

The skin and flesh color of the QGP were measured using a Konica Minolta CR-400 Chroma Meter (Konica Minolta, Osaka, Japan). Variables of color, chroma (C*), and hue angle of the peel were measured randomly at three positions along the equator of the plum, and for IF and OF, the color was measured using the puree. The average of the three values was calculated directly by the instrument and was used as a single variable value of color per plum.

#### 2.2.2. Total Soluble Solids (TSS) and Titratable Acidity (TA)

The TSS of IF and OF was determined in pureed samples using a digital (0–65%) refractometer (PR-101, ATAGO, Tokyo, Japan) [21]. TA was determined by an automatic titration system (Metrohm Dosimat 765, Karl Fischer, Metrohm, Herisau, Switzerland), using 0.2 g of the pureed IF and OF, diluted with 50 mL of distilled water. NaOH (0.1 N) was used to neutralize organic acids in the plum samples. TA was expressed as grams of malic acid equivalents per 100 g of fresh weight [21].

### 2.3. Determination of Total Phenolic Content and Analysis of Anthocyanins

#### 2.3.1. Extraction

Anthocyanins and non-anthocyanin phenolics were extracted according to Hong et al. [22] with slight modifications. Approximately 0.2 g of freeze-dried QGP was extracted with 5 mL of cold methanol-Milli-Q water (MQ-water) (80:20, *v*/*v*), 0.1 M HCl. Then, the mixture was shaken on a reciprocating shaker (RP1812, Paton Scientific, Victor Harbor, SA, Australia) for 10 min at 200 rpm/min followed by centrifugation (Eppendorf Centrifuge 5804, Eppendorf, Hamburg, Germany) at 3900 rpm for 10 min at 4 °C. The supernatant was collected, and the residue was re-extracted twice using the same procedure as described above. Finally, the supernatants were combined and filtered through a 0.2 μm membrane filter before determining the total phenolic content (TPC) and analyzing anthocyanins. All extractions were done in triplicates.

#### 2.3.2. TPC

TPC (Folin–Ciocalteu assay) was determined as reported previously [23], using a micro-plate absorbance reader (Sunrise, Tecan, Maennedorf, Switzerland) at 700 nm. TPC was expressed as milligrams of gallic acid equivalents per 100 g of sample (mg GAE/100 g).

#### 2.3.3. Anthocyanins

The analysis was carried out on an ultra-high performance liquid chromatography (UHPLC) system (Agilent, Santa Clara, CA, USA), equipped with a diode array detector (DAD), according to Hong, Netzel, and O’Hare [22] with slight modifications. The detection signal was recorded and quantified, and all anthocyanin peaks were identified by a Shimadzu liquid chromatography/tandem mass spectrometry (LC-MS/MS) Q-Trap. The detection wavelength of the DAD was set at 520 nm for quantification. The chromatographic system was equipped with a Waters BEH C18 analytical column (100 × 2.1 mm, 1.8 μm particle size; Waters, Dublin, Ireland), keeping the column temperature at 60 °C. Mobile phase A (MQ-water, acetonitrile, formic acid; 95:4:1) and mobile phase B (acetonitrile, 1% formic acid) eluted at a flow rate of 0.2 mL/min for a period of 36 min with a gradient of 100%, mobile phase A for 1 min, 15% mobile phase B for 25 min, linear increase of mobile phase B to 100% in 1 min, purging with 100% mobile phase B for 3 min, conditioning for 1 min, and re-equilibration for 5 min. Anthocyanins were quantified using an external calibration curve of cyanidin-3-glucoside (C3G).

### 2.4. Analysis of Carotenoids

#### 2.4.1. Extraction

Extraction of carotenoids from QGP was conducted as described by [24] with slight modifications. Approximately 0.5 g of freeze-dried QGP was extracted with 6 mL of 0.1% butylated hydroxytoluene (BHT) in ethanol and vortexed for 30 s. Then, 3 mL of 5% NaCl and 10 mL of 30% dichloromethane (DCM):70% n-hexane were added and vortexed for 10 s. The solution was centrifuged at 3900 rpm for 10 min at room temperature, and the top layer was removed. Samples were re-extracted twice using 10 mL of DCM:n-hexane, sonicated for 10min, and centrifuged, and then the top layer was separated. All the top layers were combined and evaporated to dryness at 35 °C using a centrifugal evaporator (DUC-23050-H00, miVac, Genevac, Ipswich, England). Dry extracts were stored at −20 °C until analysis.

#### 2.4.2. Analysis

Carotenoids were analyzed according to Calvo-Brenes, Fanning, and O’Hare [24] with modifications. On the day of analysis, samples were reconstituted with 2 mL of 50:50 methanol:methyl tert-butyl ether (MTBE), containing 0.1% BHT. Reconstituted samples were filtered using a 0.2 µm membrane filter and transferred into HPLC vials. Analysis was carried out on an Aquity UPLC system (Waters, Santa Clara, CA, USA) equipped with a Waters photodiode array (PDA) detector at 450 nm. The chromatographic system was fitted with an YMC C30 analytical column (250 × 4.6 mm, 5 μm particle size; Kinesis, Brisbane, QLD, Australia), keeping the column temperature at 23 °C, and 2 µm of each extract was injected. The gradient program of mobile phase A (0.1% formic acid in methanol (*v*/*v*)) and mobile phase B (0.1% formic acid in MTBE (*v*/*v*)) eluted at a flow rate of 0.6 mL/min for a period of 66 min with a gradient of 80% mobile phase A for 1 min, 70% mobile phase A for 25 min, 30% mobile phase A for 30 min, purging for 5 min, conditioning for 1 min, and re-equilibration for 4 min. Lutein as well as α- and β-carotene were identified by comparing their retention times, molecular masses (from a Shimadzu LC-MS/MS), and specific absorption spectra with those of commercial standards.

### 2.5. Analysis of Sugars

#### 2.5.1. Extraction

Sugars in QGP were extracted, identified and quantified according to Hong, et al. [25].

#### 2.5.2. Analysis

The sugar profiles of the QGP tissues were determined using a Shimadzu Nexera X2 UHPLC system coupled with a Shimadzu MS-8045 triple quadrupole mass spectrometer (Shimadzu, Kyoto, Japan). Briefly, the multiple reaction monitoring (MRM) mode was applied with optimal collision energy (CE) for individual sugars. The MRM transitions for fructose and glucose were m/z 179.2 → 113.1 (for quantification) and m/z 179.2 → 89.0 (for identification), for sucrose m/z 341.2 → 179.2 (for quantification) and m/z 341.2 → 161.2 → 119.1 (for identification) and sorbitol m/z 181.1 → 119.2 (for quantification) and m/z 181.1 → 113.1 → 101.0 → 89.0 (for identification).

### 2.6. Statistical Analysis

Data were subjected to factorial analysis of variance (ANOVA) for storage time and temperature using IBM SPSS Statistics for Windows, version 25.0 (IBM Corp., Armonk, NY, USA). Significant differences (*p* < 0.05) between means were determined using least significant difference (LSD). Data are presented as mean ± standard error (SE).

## 3. Results and Discussion

### 3.1. Physicochemical Properties

Changes in the physicochemical properties in different tissues of QGP are shown in Table 1. Fruit color, texture, TSS, and TA are considered as important quality characteristics of plums, which determines the harvesting maturity as well as consumer acceptance [21,26,27]. During storage at 23 °C, OF-TSS increased non-significantly (*p* > 0.05) up to 10 days of storage, and there was a non-significant (*p* > 0.05) decrease in TSS on day 14. However, at 4 °C, OF-TSS was significantly (*p* < 0.05) lower after 14 days of storage compared to day 0 (16.8% vs. 18.5%). The IF-TSS time course plots were similar to that of OF, but TSS after 14 days of storage at both temperatures were significantly (*p* < 0.05) different to day 0. The TA at 23 °C was significantly (*p* < 0.05) reduced in both OF and IF, but there was a significant (*p* < 0.05) increase at 4 °C. Organic acids are needed; they are metabolized for the respiration of plums during storage, and the metabolization is higher at high temperatures, which most likely caused the reduction in total TA [11,28]. According to Crisosto et al. [29], for plums with low TA (<1.00%), TSS:TA and TSS has a significant effect on consumer perception of sweetness, aroma, and flavor intensity of ripe plums, rather than TA.

Fruit color is an important parameter that can be used as an indicator of quality and maturity of most fruits [30]. Consumers also perceive color as an important parameter when making their purchasing decision. The color intensity of the plums is also used to estimate total anthocyanins and carotenoid levels [31]. The chroma is a measure of chromaticity (C*) and indicates the purity or saturation of color [32]. The reduction of chroma indicates an increase in the tonality of the color of the fruit [33]. There were significant differences in color parameters of the QGP tissues at both storage temperatures. Peel chroma was significantly (*p* < 0.05) reduced during the storage at 23 °C, but at 4 °C the chroma values fluctuated between storage days. Similar patterns could be observed in IF and OF chroma. Hue angles also varied between the storage days. In both IF and OF, hue angles were higher (significantly and non-significantly) at 4 °C than at 23 °C, with no clear pattern of increase or decrease during the 14 days of storage. However, according to previous studies, both peel and flesh hue angles in black/purple plums reduce significantly during ripening [21] and post-harvest storage [34], which is paralleled by the accumulation of anthocyanins, resulting in the observed color change. The identified changes in TSS, TA, and color at 4 and 23 °C can be used as “indicator” quality parameters to assess and improve the post-harvest storage conditions for plums, thus helping to preserve their palatability as long as possible. However, these properties can vary significantly depending on the maturity stage at harvest and the plum cultivar [10]. 

### 3.2. Phytochemicals

#### 3.2.1. Total Phenolics

Peel TPC was the highest (*p* < 0.05) among the three tissues of QGP (Figure 2); it was 11 times higher than that of IF and 8 times higher than that of OF on day 0. The highest TPC was recorded in the peel after 10 days of storage at 23 °C (2.3-fold increase vs. day 0).

The peel TPC in QGP was much higher than the previously reported value for ‘Black Diamond’, another red fleshed, black peel plum (1205 vs. 270 mg GAE/100 g fresh weight (FW)) [34]. Generally, the peel has a higher TPC than the flesh [7,35]. At 4 °C, the IF-TPC was higher (*p* < 0.05) than that at 23 °C, but this was the opposite in the OF-TPC after 14 days of storage. According to Díaz-Mula, Zapata, Guillén, Martínez-Romero, Castillo, Serrano, and Valero [34], the flesh TPC in ‘Black Diamond’ was 50 mg GAE/100g FW, which was lower than the QGP OF-TPC of 142 mg GAE/100 g FW. The concentration–storage-time courses of TPC and TAC (see Figure 3) in the peel samples at 23 °C are very similar. However, at 4 °C, peel-TPC continues to increase in contrast to TAC. Increase in TPC in QGP tissues during storage can be caused by the stimulation of enzymes involved in the phenolic biosynthesis pathway, such as phenylalanine ammonia-lyase, cinnamate-4-hydroxylase, and 4-coumarate coenzyme A ligase, as previously reported by others [36].

#### 3.2.2. Anthocyanins

Anthocyanins are the main pigments in Japanese plums having higher anthocyanin concentration in the peel than in the flesh [5]. The anthocyanin content of the plums varies noticeably among cultivars and continues to increase during ripening [30,37,38] and postharvest storage [11]. QGP is a dark red-fleshed and dark red-peeled (“blood”) plum; anthocyanins are accumulated in both peel and flesh, and the accumulation varies between different tissues (Figure 3). The main anthocyanins found in Japanese plums, including QGP, are cyanidin-3-O-glucoside and cyanidin-3-O-rutinoside (C3R). The TAC of the peel was highest (*p* < 0.05) after 10 days of storage at 23 °C (3-fold increase), whilst at 4 °C, the highest anthocyanin content was recorded after 14 days of storage (1.2-fold increase). Of the three tissues analyzed, peel had the highest anthocyanin content (1251 mg/100 g FW, day 10), followed by OF and IF (75 and 21 mg/100 g FW, respectively, on day 14). The peel content in the present study is considerably higher than the previously reported highest peel anthocyanin concentration of 538 mg anthocyanins/100g FW [9,10]. This can be caused by differences in the initial maturity of the fruits and/or different storage conditions.

The highest OF-TAC was observed after 14 days of storage at 4 and 23 °C and highest IF-TAC after 10 days of storage at 4 and 23 °C. There was a significant (*p* < 0.05) difference between the TAC of the peel at 4 and 23 °C at each storage day. C3G was the predominant anthocyanin found in all the tissues. There was an increase (*p* < 0.05) of C3G towards day 14 in peel (+20%), OF (+16%) and IF (+10%) at 23 °C, and peel (+4%) and IF (+17%) at 4 °C.

Flesh reddening due to the accumulation of anthocyanins has become one of the important quality parameters in red fleshed plum cultivars. In plants, anthocyanin synthesis occurs through the phenylpropanoid pathway [39]. Cold storage can stimulate the expression of specific genes involved in the biosynthesis of anthocyanins and can also enhance the activity of these enzymes in plums [40]. It was reported in the literature that exposure to chilling stress causes flesh reddening due to the rapid anthocyanin accumulation from the escalation of the phenylpropanoid pathway in different plum varieties, such as ‘Friar’ [36], ‘Sanhuali’ [40], and ‘Royal Diamond’ [41]. Interestingly, flesh reddening/anthocyanin accumulation is accelerated by moving plums from cold storage to ambient temperature [26,36], and the longer the plums were stored at low temperature, the higher the accumulation of anthocyanins at ambient temperature [36]. The QGP fruits used in this study were also stored at 2 °C for 3 weeks (common commercial practice) prior to the storage trial, and therefore it is likely that the removal from the cold storage accelerated the anthocyanin accumulation at 23 °C.

#### 3.2.3. Carotenoids

The main carotenoids identified in QGP tissues were α-carotene, β-carotene, and lutein (Figure 4). The levels of carotenoids vary between tissues. There was no difference (*p* > 0.05) between the total carotenoid content in inner and outer flesh, but there was a significant (*p* < 0.05) difference between the IF and peel and the OF and peel, respectively (Figure 5). Total carotenoid content was highest in the peel (2.52 mg/100g FW) after 14 days of storage at 4 °C. Similar results were reported for the red plum cultivars ‘Black Beaut’, ‘Red Beaut’, ‘Santa Rosa’, and ‘Angeleno’, having higher carotenoid levels in the peel than in the flesh [7]. Furthermore, there was a reduction in total carotenoids in IF and OF at both storage temperatures. Similar results were reported for dark-purple plums stored at 2 °C and subsequent storage at 20 °C [34]. β-carotene was the predominant carotenoid in all tissues analyzed, followed by α-carotene and lutein. Lutein was not detectable in most of the stored IF and OF samples, whereas the highest lutein levels were recorded in the peel (Figure 4).

### 3.3. Sugar

Different tissues have different levels of sugar (Figure 6). Total sugar content in IF and OF was higher at 4 °C than at 23 °C throughout the storage trial. There was no difference (*p* > 0.05) in total sugar in OF and IF after 14 days of storage at 4 °C, whereas a significant (*p* < 0.05) difference could be observed in the peel. At 23 °C, significant (*p* < 0.05) differences in total sugar were found in OF, IF, and peel after 14 days of storage. Peel had the highest sugar levels, followed by IF and OF.

Sorbitol, fructose, glucose and sucrose were the main sugars identified in the three different QGP tissues (Figure 6). These sugars were also found in different Japanese [6,42] and European [38] plum cultivars. On day 0, sucrose was the predominant sugar in both IF (45%) and OF (37%), whereas glucose (30%) was the main sugar in the peel. After 14 days, sucrose was still the main sugar of IF (36%, 48%) and OF (26%, 40%), respectively, at 4 and 23 °C. It was noted that after 14 days of storage at 4 °C, the sucrose content in all tissues decreased, while there was a parallel increase in glucose and fructose (Figure 6). The decrease of sucrose and increase in glucose and fructose levels could be a result of the hydrolysis of sucrose to glucose and fructose. Similar results of reduction in sucrose could be observed in a 5-week storage trial with ‘Amber Jewel’ plums at 0 and 5 °C. However, in the same experiment for ‘Blackamber’ plum, there was an increase in the sucrose accumulation during storage at 0 °C for 5 weeks [42]. After 14 days of storage at 23 °C, there was a significant (*p* < 0.05) increase in sucrose in the peel (+25%), and a significant (*p* < 0.05) decrease in OF (−12%) and IF (−17%). These variations suggest that there can be differences in the sugar accumulation and metabolism in plums depending on the cultivar or genotypic [38,42], storage conditions [42], and tissue. However, this needs to be investigated further. After 14 days of storage at 23 °C, there was a significant (*p* < 0.05) reduction in sorbitol in all tissues, whereas at 4 °C, sorbitol was only reduced (*p* < 0.05) in peel and IF but slightly (*p* > 0.05) increased (+7%) in OF. Sorbitol is a sugar alcohol that can be used as a natural replacement for sucrose and glucose in special diets for diabetic people [43].

## 4. Conclusions

The effect of storage temperature (4 and 23 °C) and storage time (up to 14 days) on physicochemical properties, phytochemicals, and sugar composition in different tissues of QGP was demonstrated in the present study. The increase in bioactive anthocyanins in flesh and peel was more prominent at ambient temperature (23 °C) than at refrigerated (4 °C) storage. There was also a decrease in total sugars and TA at ambient storage, which may affect the sensory attributes of QGP. However, the actual impact of this on the overall fruit quality and consumer acceptance needs to be investigated further.

## Figures and Tables

**Figure 1 foods-10-00352-f001:**
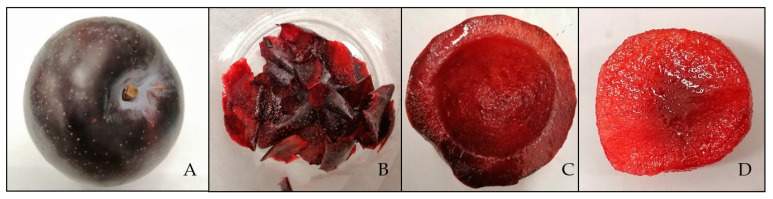
(**A**) Mature Queen Garnet plum (QGP); (**B**) Peel; (**C**) Outer flesh (OF); (**D**) Inner flesh (IF).

**Figure 2 foods-10-00352-f002:**
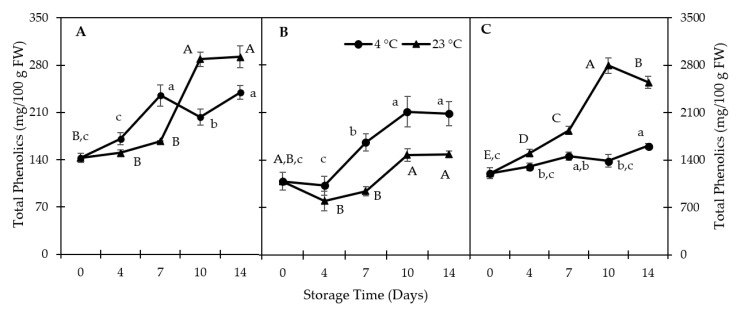
Total phenolic content (TPC) in (**A**) outer flesh; (**B**) inner flesh; (**C**) peel. Data are presented as mean ± SE (*n* = 6–10). Different uppercase and lowercase letters indicate significant (*p* < 0.05) differences between storage days; FW-fresh weight

**Figure 3 foods-10-00352-f003:**
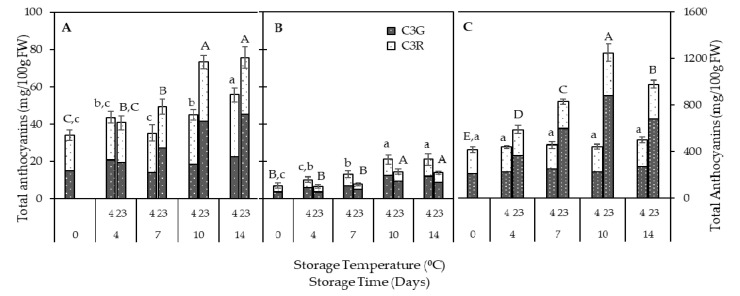
Anthocyanins in (**A**) outer flesh; (**B**) inner flesh; (**C**) peel. Data are presented as mean ± SE (*n* = 6–10). Different uppercase letters indicate significant (*p* < 0.05) differences between storage days at 23 °C, and different lowercase letters indicate significant (*p* < 0.05) differences between storage days at 4 °C; C3G-cyanidin-3-O-glucoside; C3R-cyanidin-3-O-rutinoside.

**Figure 4 foods-10-00352-f004:**
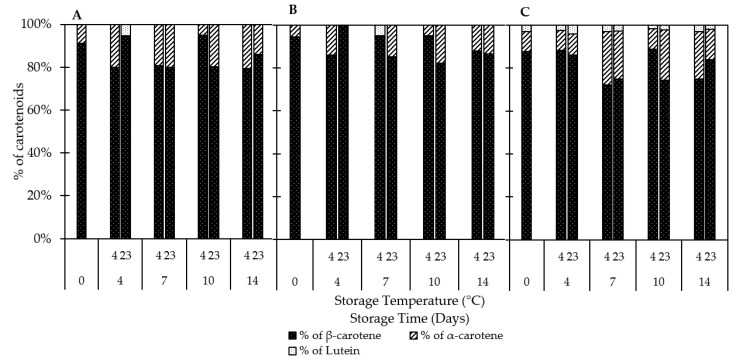
Individual carotenoids in (**A**) outer flesh; (**B**) inner flesh; (**C**) peel. Data are presented as mean (SE not shown; *n* = 3).

**Figure 5 foods-10-00352-f005:**
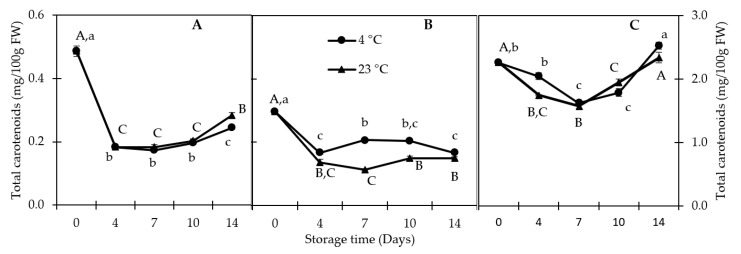
Carotenoids in (**A**) outer flesh; (**B**) inner flesh; (**C**) peel. Data are presented as mean ± SE (*n* = 3). Different uppercase and lowercase letters indicate significant (*p* < 0.05) differences between storage days.

**Figure 6 foods-10-00352-f006:**
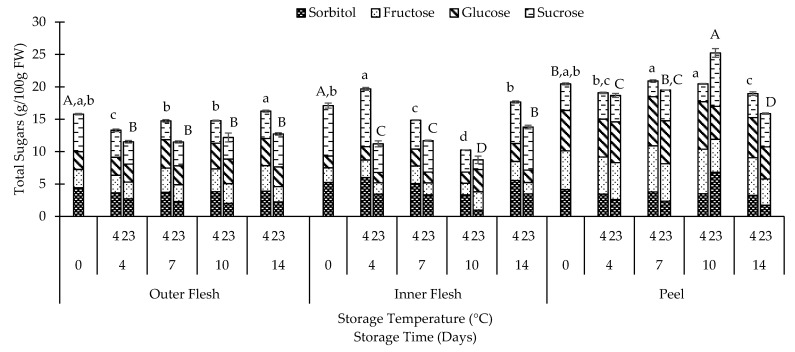
Total sugars (mean ± SE) and individual sugars (mean) in different QGP tissues (*n* = 3). Different uppercase letters indicate significant (*p* < 0.05) differences between storage days at 23 °C, and different lowercase letters indicate significant (*p* < 0.05) differences between storage days at 4 °C.

**Table 1 foods-10-00352-t001:** Color (chroma and hue angle), total soluble solids (TSS; %), and titratable acidity (TA; as % of malic acid) of the peel, outer flesh (OF) and inner flesh (IF) of Queen Garnet plum (QGP) stored at 4 and 23 °C for 14 days.

Physicochemical Property	Storage Temperature (°C)	Storage Time (Days)				
		0	4	7	10	14
Peel chroma	4	11.35 ± 0.80 a,b,A	12.56 ± 0.77 a,A	11.89 ± 0.92 a,A	9.82 ± 0.33 b,A	11.25 ± 0.68 a,b,A
	23	11.35 ± 0.80 a,A	7.85 ± 0.43 b,B	5.87 ± 0.49 c,B	4.82 ± 0.33 c,d,B	3.58 ± 0.31 d,B
OF chroma	4	15.44 ± 0.83 b,A	18.93 ± 0.97 a,A	22.15 ± 0.77 a,A	21.41 ± 1.69 a,A	19.48 ± 1. 26 a,A
	23	15.44 ± 0.83 a,b,A	15.78 ± 1.57 a,b,A	18.21 ± 1.17 a,B	12.86 ± 0.96 b,c,B	10.46 ± 0.96 c,B
IF chroma	4	18.97 ± 1.15 b,A	23.34 ± 0.62 a,A	25.40 ± 1.29 a,A	24.48 ± 1.45 a,A	24.79 ± 1.21 a,A
	23	18.97 ± 1.15 a,A	19.09 ± 1.34 a,B	19.05 ± 1.29 a,B	17.70 ± 1.20 a,B	16.69 ± 1.18 a,B
Peel hue	4	9.28 ± 1.30 a,b,A	12.06 ± 1.18 a,A	7.21 ± 1.33 b,A	6.32 ± 0.46 b,c,A	8.56 ± 1.10 a,b,A
	23	9.28 ± 1.30 b,c,A	7.75 ± 1.68 c,B	14.58 ± 1.84 a,B	12.03 ± 1.15 a,b,B	14.21 ± 1.42 a,B
OF hue	4	17.83 ± 0.39 c,A	18.41 ± 0.65 b,c,A	20.55 ± 0.56 a,A	19.82 ± 0.73 a,b,A	19.94 ± 0.06 a,b,A
	23	17.83 ± 0.39 a,A	17.72 ± 2.79 a,A	15.60 ± 0.54 b,B	13.57 ± 0.49 d,B	15.17 ± 0.84 b,c,d,A
IF hue	4	31.16 ± 3.70 a,A	27.19 ± 2.29 a,bA	24.20 ± 1.68 b,A	21.93 ± 0.72 b,A	22.72 ± 1.18 b,A
	23	31.16 ± 3.70 a,A	25.13 ± 3.04 a,A	21.29 ± 0.89 b,A	15.54 ± 1.72 b,c,B	13.33 ± 0.76 c,B
OF TSS	4	18.5 ± 0.2 a,A	18.2 ± 0.3 a,b,A	16.7 ± 0.4 c,A	17.3 ± 0.5 b,c,A	16.8 ± 0.4 c,A
	23	18.5 ± 0.2 a,A	18.5 ± 0.4 a,A	18.7 ± 0.3 a,B	18.7 ± 0.3 a,B	17.8 ± 0.4 a,A
IF TSS	4	18.8 ± 0.2 a,A	18.7 ± 0.7 a,A	17.4 ± 0.4 b,A	17.9 ± 0.4 b,A	17.4 ± 0.3 b,A
	23	18.8 ± 0.2 a,A	18.6 ± 0.3 a,A	18.8 ± 0.3 a,B	18.5 ± 0.2 a,A	17.5 ± 0.3 b,A
OF TA	4	0.74 ± 0.05 c,A	0.75 ± 0.02 c,b,A	0.89 ± 0.04 a,A	0.87 ± 0.03 a,A	0.89 ± 0.04 a,A
	23	0.74 ± 0.05 a,A	0.70 ± 0.05 a,A	0.70 ± 0.03 a,B	0.63 ± 0.03 a,b,B	0.54 ± 0.00 b,B
IF TA	4	0.46 ± 0.04 c,A	0.56 ± 0.03 b,A	0.68 ± 0.02 a,A	0.60 ± 0.03 b,A	0.61 ± 0.03 a,b,A
	23	0.46 ± 0.04 a,b,A	0.51 ± 0.06 a,A	0.41 ± 0.02 c,b,B	0.39 ± 0.02 c,b,B	0.36 ± 0.02 c,B

Data are presented as mean ± standard error (SE), with *n* = 6–10. For each parameter, different uppercase letters show significant (*p* < 0.05, least significant difference (LSD)) differences between the two storage temperatures (4 and 23 °C) for each storage day, and different lowercase letters within each row show significant (*p* < 0.05) differences between storage days.

## Data Availability

Data is contained within the article.

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
