# Peer review of "Effect of Storage on the Nutritional Quality of Queen Garnet Plum"

_foods, 2021, doi:10.3390/foods10020352_

Round 1

Reviewer 1 Report

In my opinion, this paper deals with an interesting topic i.e., “Effect of storage on the nutritional quality of Queen Garnet plum”. The topic is interesting and the work well written and organised, and the title coherent with content of paper.

As well, also the methods are clear, well presented, and organized. Although uncommon instrumental methods are used for the determination of routine compounds (ex., sugars), they help to understand and follow the results. These are well supported by data and literature. The statistical study of the obtained data is correct, and the final conclusions are promising (quality parameters to assess and improve the post-harvest storage conditions for plums preserving their palatability if possible).

I suggest revising the manuscript according to the following comments.

Page 2, line 86

Authors could explain the sentence “unless otherwise stated”.

Page 2, line 106

The sentence “cold aqueous methanol 80% with 0.1 M HCl” is confusing. Is it a solution of 0.1 M HCl in MeOH?

Page 3, line 126

Do the authors have verified that the temperature of 60 °C does not influence the stability of the anthocyanins during the analysis?

Page 3, line 131

The authors could specify the total time of analysis.

Page 4, line 145

In the paragraph 2.4.2, can the authors add the analytical conditions?

Page 4, line 157

In the paragraph 2.5.1, the description of Extraction method for sugars analysis, in my opinion, is unnecessary if the authors report the reference.

Page 4, lines 184-186

The authors could change the sentence in "During storage at 23 °C, OF-TSS increased non-significantly (p>0.05) up to 14 days of storage (not 10 days as the authors affirm) and there was not a slight significant reduction in TSS on day 14.

Page 4, line 187

The sentence in not clear. The number 16.8% expresses the observed decreasing o is it the TSS value?

Page 5, line 195

I ask to the authors to explore more the affirmation “TA does not have any significant correlation with these sensory attributes (i.e., sweetness). I Know that sweetness of some fruit influenced by acids and then (AT) as like as sugars/organic acids ratio.

Page 7, line 244

In my opinion, the authors refer to figure 3, not figure 2. Could the authors verify this?

Page 8, line 281

In the figure 4. In my opinion, there aren't difference regarding total carotenoid content in the peel (2.52 mg/100g FW) at different temperature after 14 days. In this case, does not the temperature influence the content. In addition, I saw (figure 4) that total carotenoid content was highest in the peel after 14 days of storage at 4°C, not at 23°C as the authors affirm.

Page 10, lines 319-321

Can the authors verify the reported results in the sentence “After 14 days of storage at 23 °C, there was a significant (p<0.05) reduction in sorbitol in all tissues, whereas at 4 °C, sorbitol was only reduced (p<0.05) in peel and IF, but slightly (p>0.05) increased (+7%) in OF”? In my opinion the sorbitol increased slightly increased in IF and not in OF.

Table 1.

In the legend: the authors could eliminate the parenthesis unnecessary. In addition, I suggest adding one decimal for the all OF-TSS and IF-TSS values as done for other ones.

Figure 3.

It is not easily understood because it is not clear what specific difference the upper- and lower-case letters refer to. Do capital letters refer to differences between different temperatures at each storage time, or difference between samples at a certain temperature during different storage times? Indicate this information in the legend. This could improve the understanding of the figure. Same considerations for figures 5 and 6.

Reviewer 2 Report

In this work the authors carry out an exhaustive study on the most suitable storage conditions for plums, specifically the 'Queen Garnet' variety. This information will undoubtedly be of great use to the population and could extend the quality of plums and thus their properties, so I suggest following changes:

1.-The objective of this study was to determine the impact of common domestic storage temperatures (4 ° C and 23 ° C). However, it would have been interesting to include information at an intermediate temperature.

2.- Why was such a high temperature of 60ºC used in the anthocyanin determination? If it is an established method include the reference

3.- I also suggest reducing the number of references in the article from 60 to 30-40.

4.- The increase in bioactive anthocyanins in meat and skin was more prominent at room temperature (23 ° C) than in refrigerated storage at 328 (4 ° C) which from my point of view is the most useful conclusion of this study . However, many methodologies were slightly modified with respect to the original ones, my question is whether the authors validated them after these modifications. This aspect, which undoubtedly affects the absolute value of the results, is of vital importance and should be clarified in the text.

Round 2

Reviewer 2 Report

I recommend the acceptance of the article in this form. The results are very interesting for readers.